# Taxonomic and functional turnover are decoupled in European peat bogs

Bjorn J.M. Robroek [1,2], Vincent E.J. Jassey[3], Richard J. Payne[4,5], Magalí Martí[6], Luca Bragazza[7,8,9], Albert Bleeker [10], Alexandre Buttler[7,8], Simon J.M. Caporn[4], Nancy B. Dise[4,11], Jens Kattge [12,13], Katarzyna Zając[14], Bo H. Svensson[6], Jasper van Ruijven[15] & Jos T.A. Verhoeven[1]

In peatland ecosystems, plant communities mediate a globally significant carbon store. The effects of global environmental change on plant assemblages are expected to be a factor in determining how ecosystem functions such as carbon uptake will respond. Using vegetation data from 56 *Sphagnum*-dominated peat bogs across Europe, we show that in these ecosystems plant species aggregate into two major clusters that are each defined by shared response to environmental conditions. Across environmental gradients, we find significant taxonomic turnover in both clusters. However, functional identity and functional redundancy of the community as a whole remain unchanged. This strongly suggests that in peat bogs, species turnover across environmental gradients is restricted to functionally similar species. Our results demonstrate that plant taxonomic and functional turnover are decoupled, which may allow these peat bogs to maintain ecosystem functioning when subject to future environmental change.

[1] Ecology and Biodiversity, Department of Biology, Utrecht University, Padualaan 8, NL-3584 CH Utrecht, The Netherlands. [2] Biological Sciences, Faculty of Natural and Environmental Sciences, Institute for Life Sciences, University of Southampton, Southampton SO17 1BJ, UK. [3] Université de Toulouse, INP, UPS, CNRS, Laboratoire d'Ecologie Fonctionnelle et Environnement (Ecolab), 118 Route de Narbonne, 31062 Toulouse Cedex, France. [4] School of Science and the Environment, Division of Biology and Conservation Ecology, Manchester Metropolitan University, Manchester M1 5GD, UK. [5] Environment, University of York, Heslington, York YO10 5DD, UK. [6] Department of Thematic Studies–Environmental Change, Linköping University, SE-581 83 Linköping, Sweden. [7] Department of Life Science and Biotechnologies, University of Ferrara, Corso Ercole I d'Este 32, I-44121 Ferrara, Italy. [8] École Polytechnique Fédérale de Lausanne (EPFL), Ecological Systems Laboratory (ECOS), CH-1015 Lausanne, Switzerland. [9] WSL - Swiss Federal Institute for Forest, Snow and Landscape Research, Site Lausanne, CH-1015 Lausanne, Switzerland. [10] Unit Water, Agriculture and Food, PBL Netherlands Environmental Assessment Agency, PO Box 30314, NL-2500 GH The Hague, The Netherlands. [11] Centre for Ecology and Hydrology, Edinburgh Bush Estate, Penicuik EH26 0QB Edinburgh, UK. [12] Max Planck Institute for Biogeochemistry, Hans Knöll Straße 10, D-07745 Jena, Germany. [13] German Centre for Integrative Biodiversity Research (iDiv), Halle-Jena-Leipzig, Deutscher Platz 5e, D-04103 Leipzig, Germany. [14] Limnological Research Station and Department of Hydrology, University of Bayreuth, Universitätsstraße 30, D-95440 Bayreuth, Germany. [15] Plant Ecology and Nature Conservation, Wageningen University and Research Centre, PO Box 47, NL-6700 AA Wageningen, The Netherlands. Bjorn J.M. Robroek and Vincent E.J. Jassey contributed equally to this work. Correspondence and requests for materials should be addressed to B.J.M.R. (email: bjorn.robroek@soton.ac.uk)

Plant community composition plays an important role in regulating ecosystem processes[1, 2] with mounting evidence that high diversity safeguards ecosystem functioning in a changing environment[3]. Anthropogenic drivers of environmental change such as increased temperature, drought or nutrient deposition can, however, erode diversity[4–8] and alter the composition of plant communities[9]. Environmental change could even lead to the emergence of novel assemblages[10, 11]. Changes in species composition are often coupled to changes in functional trait composition[12]. As functional trait composition is generally assumed to be an important determinant of ecosystem services[13], shifts in plant community composition may impact ecosystem service provision[12, 14]. The effects of environmental change on diversity and community composition, however, largely depend on the nature of biotic changes[15, 16] and are ecosystem[5]—and scale—dependent[17], and therefore difficult to generalise.

Environmental filtering has been mathematically demonstrated to lead to species assemblages with groups of species that are similar in traits, while traits between groups are divergent[18, 19]. One possibility is that species composition and community—trait composition could become uncoupled under rapid environmental change[20]. A convergence in trait composition may lead to a decline in functional diversity, which then is expected to negatively affect ecosystem functioning[21, 22]. As such, studies which aim to identify the effect of environmental change should not only focus on plant taxonomic diversity, but should also consider the functional composition of the plant communities[23]. This is important as the magnitude of change in ecosystem functioning may strongly depend on changes in the functional identity of species in the community. If, for example, species share functional traits (i.e., they are functionally redundant), the potential consequences of species loss (or gain) for ecosystem processes are likely to be minimal. If, on the other hand, species with unique traits are lost from the community (or gained), important functions of the ecosystem may be affected[24, 25]. As the ability of ecosystems to maintain important functions depends on community characteristics such as species richness and functional trait diversity, it is critical to improve our knowledge of how environmental change relates to the taxonomic and trait composition of plant communities.

Northern peatlands are an important component of the global carbon (C) cycle, as they represent a large but vulnerable pool of soil carbon. Globally, peatlands contain over 500 billion tonnes of C in just 3% of the Earth's land surface[26], which is 16% of all C in terrestrial ecosystems[27] and 67% of the carbon in the atmosphere[26]. Peatland plant species composition is central to how peatlands will respond to environmental change given the strong links between plant community composition and ecosystem processes such as C cycling[28–31]. While the composition of peatland plant communities has long been described as remarkably stable[32, 33], progressive changes in environmental conditions may cause shifts in the relative abundance of species[6]. Combined

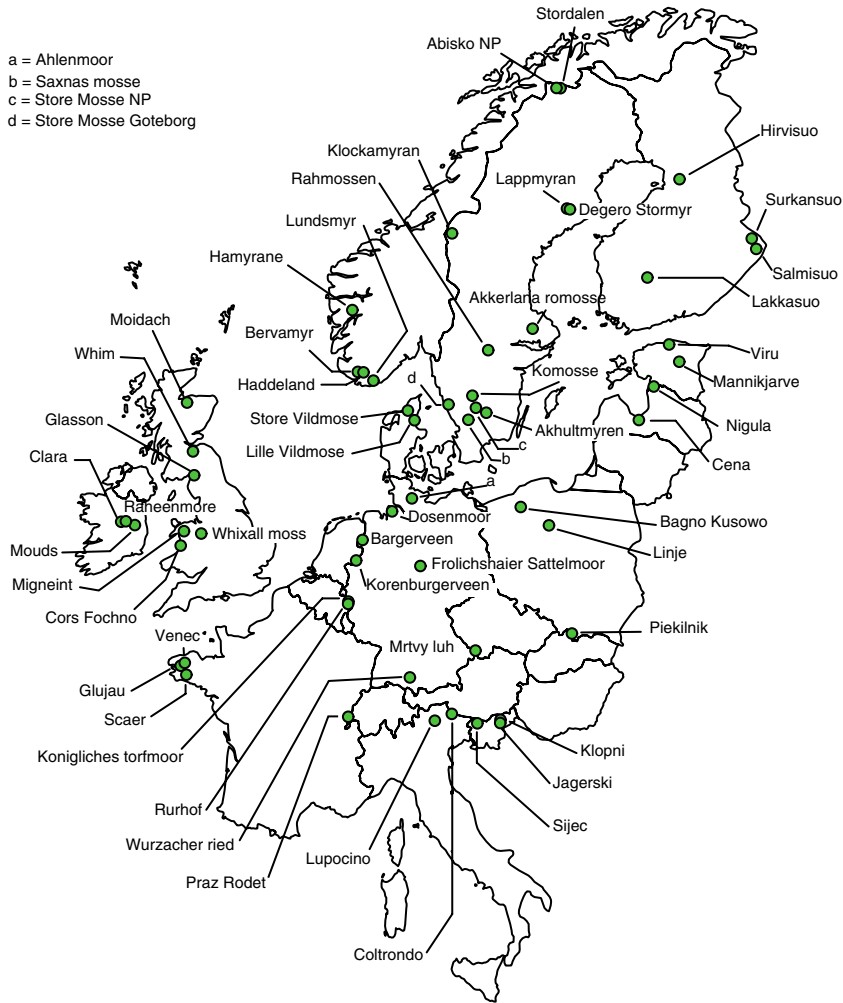

**Fig. 1** Geographic locations of the 56 European peat bogs. Distribution of the sampled peat bogs across Europe. Map image source: R package *rworldmap*

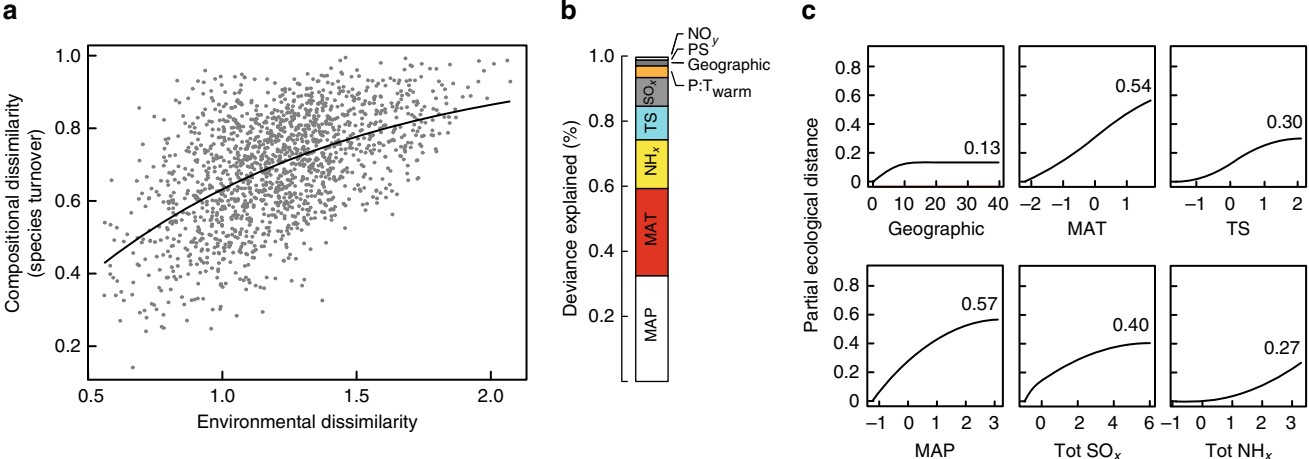

**Fig. 2** Taxonomic turnover along environmental gradients. **a** Relationship between observed plant community compositional dissimilarity between site pairs (species turnover or β-diversity) and their predicted environmental dissimilarity. The line represents the linear predictor of the regression equation from generalized dissimilarity modelling (GDM, Methods section). **b** Reduction in deviance explained between full model and model with the environmental variable omitted, i.e., an indicator of the proportion of deviance attributed to that variable. Variables tested were geographical distance, mean annual temperature (MAT), seasonality in temperature (TS), mean annual precipitations (MAP), seasonality in precipitation (PS), ratio of precipitation and temperature of the warmest quarter (P:$T_{warm}$), sulphate ($SO_x$), and reduced ($NH_x$) and oxidised ($NO_x$) nitrogen atmospheric depositions. **c** Partial regression fits (Model-fitted-I-splines) for variables significantly associated with plant community species turnover. Note that we also included geographical distance for reference. The maximum height (inset number) reached by each I-spline curve indicates the relative importance of that variable in explaining beta diversity, keeping all other variables constant (i.e., the partial response curve value). The shape of each function provides an indication of how the rate of compositional turnover varies along the environmental gradient

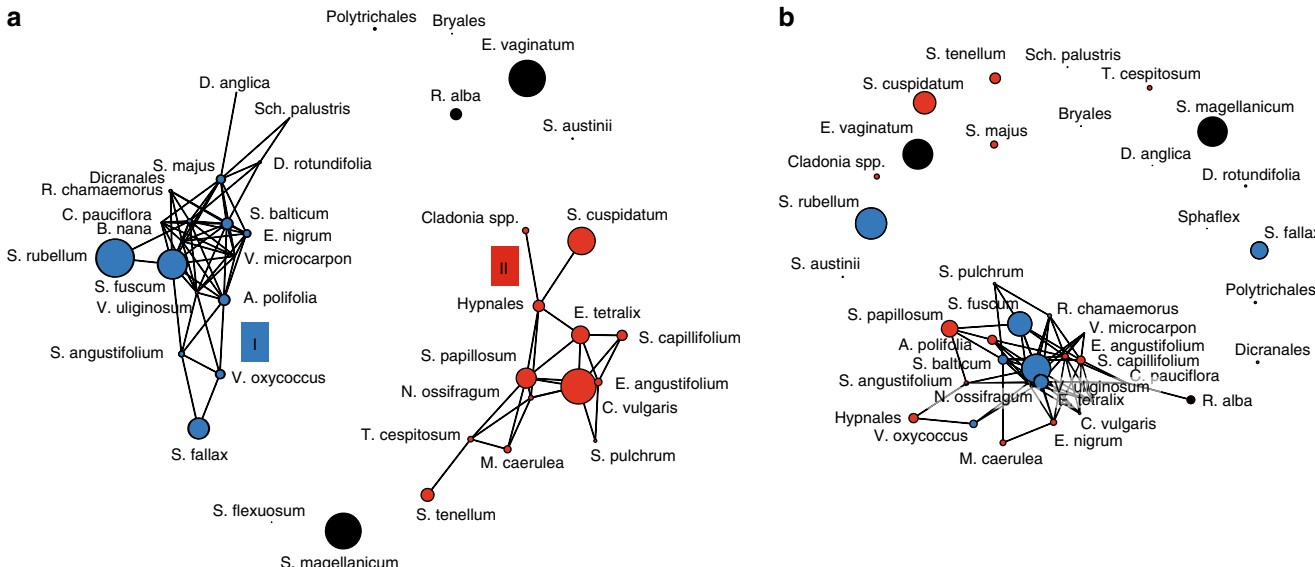

**Fig. 3** Co-response clusters representing species similar in abundance-environment relationships. **a** Positive species interactions. **b** Negative species interactions. Colours represent two distinct co-response clusters (blue = I; red = II), while non-connected (black) points represent cluster-unrelated species. Black lines indicate correlations between species with a correlation coefficient ≥ 0.6. Circle size indicates the averaged abundance of species among all sites. See Supplementary Table 3 for full taxonomic names

with the direct effects of environmental change[26], shifts in plant community composition could amplify the effects of environmental change on the peatland carbon balance[34].

Here we present an analysis of the relationships between plant species composition, functional trait composition, and a number of major environmental drivers, such as temperature, precipitation and atmospheric deposition, using data on plant species composition from 56 ombrotrophic *Sphagnum*-dominated peat bogs across Europe (Fig. 1). We provide evidence for general

patterns in species turnover (i.e., change in species composition, or species replacement) along environmental gradients: peat bog plant species are divided into two clusters with convergent within–cluster, but divergent between–cluster responses along the main environmental gradients. We conclude that non-random species replacement of functionally identical species between both networks conserves functional redundancy at the ecosystem level, which may sustain the robustness of peat bog ecosystems to anthropogenic drivers of environmental change.

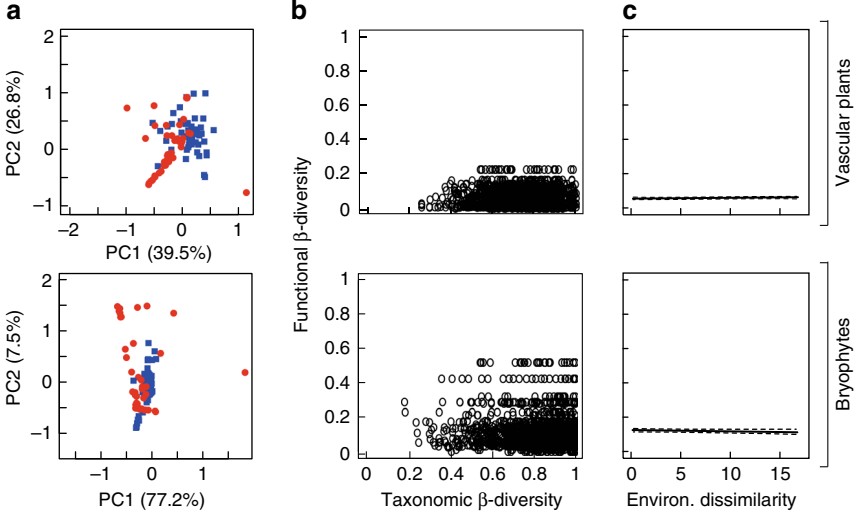

**Fig. 4** Functional community composition and functional turnover. **a** Functional community composition of the two co-response clusters (blue squares = cluster I, red circles = cluster II). Principal component analysis of cluster-weighted means of the five quantitative vascular plant traits and eight bryophyte traits (see Supplementary Fig. 4 for a list of traits, and their cluster-weighted means). **b** Relationship between taxonomic and functional turnover of the vascular plant (top) and bryophyte (bottom) communities. **c** Relationship between environmental dissimilarity and functional turnover (vascular plants: $F_{1,3078} = 1.3$, $r^2 = 0.02$, $P = 0.23$; bryophytes: $F_{1,3078} = 1.4$, $r^2 = 0.02$, $P = 0.26$; Generalized Linear Model)

## Results

**Diversity indices and species turnover.** Peatland plant species richness was highly variable across the studied sites and ranged from 9 species in northwest France (Glujeau, Fig. 1) to 32 species in southwest Norway (Håmyrane, Fig. 1). Stepwise multiple regression analysis identified four variables that together explained 38% of the variability in species richness ($F_{5,50} = 7.8$, $P \leq 0.001$; stepwise multiple regression), of which latitude was the most important (Supplementary Table 1). Species richness increased with moisture index (P:T$_{warm}$) and decreased with mean annual temperature. Diversity (Simpson) showed similar patterns ($F_{7,48} = 2.6$, $P = 0.024$; stepwise multiple regression), except that it was negatively related to seasonality in precipitation (PS) and total oxidised nitrogen deposition (Supplementary Table 1).

Patterns in plant species turnover (β-diversity) along the environmental gradients were analysed using generalised dissimilarity modelling (GDM, ref. [35]). Plant species turnover (i.e., community compositional dissimilarity) increased with environmental dissimilarity (Fig. 2a). Geographic distance (i.e., latitude, longitude) contributed little to species turnover (Fig. 2b, c). Species turnover was significantly related to three climatic factors (mean annual temperature, mean annual precipitation and, to a lesser extent, seasonality in temperature) and two atmospheric deposition variables (SO$_x$ and NH$_x$) (Fig. 2c). GDM results are consistent with alternative community-level models for the identification of environmental variables linked to species turnover (Supplementary Fig. 1).

**Plant species group in distinct clusters.** We identified two major clusters of peatland species with shared within-cluster responses (Fig. 3a, Supplementary Fig. 2), but opposite between-cluster responses to environmental conditions (Fig. 3b). Both clusters were composed of common peat bog species such as *Andromeda polifolia*, *Drosera rotundifolia*, *Sphagnum fallax* and *S. rubellum* in Cluster I, and *Calluna vulgaris*, *Eriophorum angustifolium*, *Sphagnum cuspidatum* and *S. papillosum* in Cluster II. Cluster I also included some species with a more northern distribution (e.g., *Betula nana*, *Empetrum nigrum*, *Rubus chamaemorus*, *S.*

*balticum*). For a detailed view and list of cluster-associated species, see Fig. 3 and Supplementary Table 2.

Similar to species turnover (Fig. 2), cluster-specific species responses were mainly related to mean annual temperature (MAT), temperature seasonality (TS), and mean annual precipitation (MAP). Cluster I species decreased with higher temperature and precipitation, but increased with greater seasonality in temperature. Opposite responses were found for cluster II species. Increasing atmospheric SO$_x$ and NH$_x$ deposition negatively affected cluster I species, but did not, or only weakly, affect the probability of occurrence of cluster II species (Supplementary Fig. 2). We did not observe significant residual correlations between species (Supplementary Fig. 3), suggesting that biotic interactions between species were of little importance for species occurrence along the gradients.

Principal component analyses showed that the two clusters overlapped in their functional composition for both the vascular plant and bryophyte communities (Fig. 4a). However, some differences were apparent between the two clusters (Supplementary Fig. 4). Cluster I vascular plant species were smaller, produced less seeds and had higher specific leaf area than cluster II species. Cluster I bryophytes were taller, with smaller spores, narrower stem leaves, lower tissue N and higher tissue P than cluster II bryophytes (Supplementary Fig. 4).

**Functional turnover and redundancy.** Functional turnover (i.e., the change in functional composition of the community) in the vascular plant and bryophyte communities was weakly related to taxonomic turnover (vascular plant: $F_{1,3078} = 0.71$, $r^2 = 0.01$, $P = 0.41$; bryophytes: $F_{1,3078} = 1.1$, $r^2 = 0.02$, $P = 0.29$; Generalized Linear Model; Fig. 4b). Indeed, in contrast to taxonomic composition (Fig. 1b), functional turnover in both communities was low, and not affected by changing environmental conditions (i.e., environmental dissimilarity; Fig. 4c). Functional redundancy (FR, i.e., the ability of the community to maintain its function, inferred from multiple traits) of the vascular plant and bryophyte communities remained stable over climatic and atmospheric deposition gradients (Fig. 5). In contrast, functional redundancy of the two clusters changed along the environmental gradients (Fig. 5,

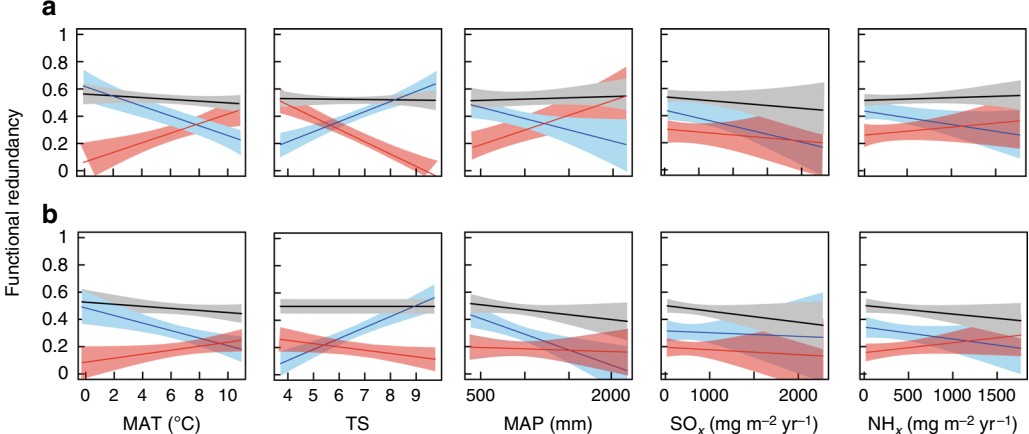

**Fig. 5** Relationships between the major environmental variables and functional redundancy. Relationships for the vascular plant community **a** and bryophyte community **b**. Black lines represent the whole plant community (black), while the coloured lines represent the plant species in the two co-response clusters (blue = cluster I, red = cluster II). MAT = mean annual temperature (°C), TS = seasonality in temperature (°C), MAP = mean annual precipitation (mm), $SO_x$ = total sulphur deposition (mg m$^{-2}$ yr$^{-1}$), $NH_x$ = total reduced nitrogen deposition (mg m$^{-2}$ yr$^{-1}$). Linear regression models and two-tailed $P$-values are shown in Table 1

Table 1). These patterns were similar for vascular plants and bryophytes, except for mean annual precipitation (MAP), which was only positively correlated with FR of vascular plants in cluster II. Atmospheric deposition was not significantly related to FR (Fig. 5, Table 1).

**Impact of changed species assembly on functional redundancy.** Reshuffling the species (NULL1) or the relative abundance of co-occurring species (NULL2) did not affect functional redundancy (FR) of the vascular plant and bryophyte communities (SES ≈ 0, Fig. 6). Random assembly of species reduced FR of both vascular plants and mosses (RANDOM). A similar reduction of FR was observed with species loss from cluster I (LOSS1). The strongest reduction in FR occurred when species from cluster II (LOSS2) were lost. Reductions in FR were also observed when species of one of the clusters were replaced by randomly selected species from the other cluster (TURNOVER1 and TURNOVER2), and was highest when cluster II species were replaced by random species of cluster I (Fig. 6).

**Discussion**
Our results show clear patterns of species turnover in European peat bog ecosystems across environmental gradients, in particular those associated with climate. Our analyses demonstrate that the plant species can be grouped into two clusters which show a high degree of similarity in functional structure. Species within a cluster are very similar in their response to environmental variables, whilst species from different clusters respond oppositely to environmental variables. Despite significant species turnover, the functional composition (i.e., cluster-weighted mean traits) of the communities as a whole remained largely unaffected by changes in environmental conditions. Taken together, our results demonstrate that environmental filtering acts primarily on the taxonomic composition of the peat bog vegetation. These findings suggest that species turnover is associated with deterministic replacement of functionally similar species. Such apparent decoupling of taxonomic and functional turnover may be an important mechanism underlying the capability of peat bog ecosystems to maintain functioning under environmental change.

Responses of plants to environmental changes are often considered to be species-specific, yet we provide evidence for common patterns in species turnover along environmental gradients.

The effects of environmental factors on species richness were minor, which is consistent with a recent comparison of a variety of ecosystems where environment-species richness relationships were least pronounced in peat bogs[5]. We did, however, observe clear effects of environmental change on peat bog species composition. Temperature, precipitation and atmospheric deposition (nitrogen and sulphate) were particularly important in explaining patterns in species turnover. These patterns seem to result from cluster-specific species responses to environmental conditions. Moreover, peat bog plants aggregate into two co-occurrence clusters with opposite responses to environmental factors, whilst species from different clusters are subject to contrasting environmental filters, conforming with the theory of community 'response rules'[36]. The existence of groups of species with distinct responses along environmental gradients echoes theory that natural communities tend to self-organise into groups of similar species along niche axes[18].

The key question is what are the consequences of the relationships between environmental factors and species turnover for the functioning of the ecosystem[37]. In our study, taxonomic turnover did not strongly relate to a turnover in the functional composition inferred from the multiple traits used in this analysis. This finding may have been determined by our choice of plant functional traits[38, 39]. Yet, random trait removal indicates that trait identity and number of traits did not affect the values of functional diversity and redundancy (Supplementary Fig. 5). The use of averaged trait values derived from trait data bases may not always accurately reflect trait values directly measured on-site[40], but has been shown to be effective for addressing questions of trait-environment relationships[41]. Although we generalise from a limited set of traits available for all the species in our communities, our findings suggest that whilst changing environmental conditions alter the species composition of European *Sphagnum*-dominated peat bogs, their functional composition remains rather unaffected. Our data provide empirical evidence that, within the studied environmental constraints in which peatland plants can exist, peatland plant community taxonomic and functional composition are decoupled.

Based on the traits used in our analysis, which are common and widely used for vascular plants and for *Sphagnum* mosses, net community functional redundancy along the environmental gradients remained remarkably unaffected. These results suggest that species replacement along environmental gradients are non-

**Table 1 Relationships between functional redundancy and environment**

|  |  | Vascular plants | | | Bryophytes | | |
|---|---|---|---|---|---|---|---|
|  |  | *F*-value | *P*-value | *r* | *F*-value | *P*-value | *r* |
| MAT | Plant community | 1.3 | 0.26 | −0.07 | 1.5 | 0.23 | −0.09 |
|  | Cluster I | **19.3** | **≤0.001** | **−0.51** | **10.9** | **≤0.01** | **−0.41** |
|  | Cluster II | **11.4** | **≤0.001** | **0.41** | 3.7 | 0.06 | 0.22 |
| TS | Plant community | 0.0 | 0.83 | 0.13 | **6.2** | **≤0.01** | **0.29** |
|  | Cluster I | **29.0** | **≤0.001** | **0.59** | **34.6** | **≤0.001** | **0.63** |
|  | Cluster II | **39.6** | **≤0.001** | **−0.66** | **4.1** | **0.04** | **−0.24** |
| MAP | Plant community | 0.2 | 0.68 | 0.12 | 1.8 | 0.18 | −0.12 |
|  | Cluster I | **5.1** | **≤0.05** | **−0.27** | **11.0** | **≤0.001** | **−0.40** |
|  | Cluster II | **7.2** | **≤0.01** | **0.33** | 0.1 | 0.74 | 0.14 |
| SO$_x$ | Plant community | 0.7 | 0.39 | −0.06 | 0.2 | 0.64 | −0.12 |
|  | Cluster I | 2.2 | 0.14 | −0.15 | 0.8 | 0.86 | −0.13 |
|  | Cluster II | 0.2 | 0.63 | −0.12 | 0.2 | 0.68 | −0.13 |
| NH$_x$ | Plant community | 0.3 | 0.59 | −0.11 | 1.9 | 0.18 | −0.13 |
|  | Cluster I | 2.6 | 0.11 | −0.17 | 1.7 | 0.19 | −0.12 |
|  | Cluster II | 0.7 | 0.39 | −0.07 | 1.9 | 0.17 | 0.13 |

Summary of generalised linear models testing the relationships between functional redundancy of vascular plants and bryophytes, and bioclimatic variables: MAT mean annual temperature, TS seasonality in temperature, MAP mean annual precipitations, SO$_x$ sulphate, and NH$_x$ nitrogen atmospheric depositions. Linear regressions were made for overall community, Cluster I species and Cluster II species. Significant correlations are in bold

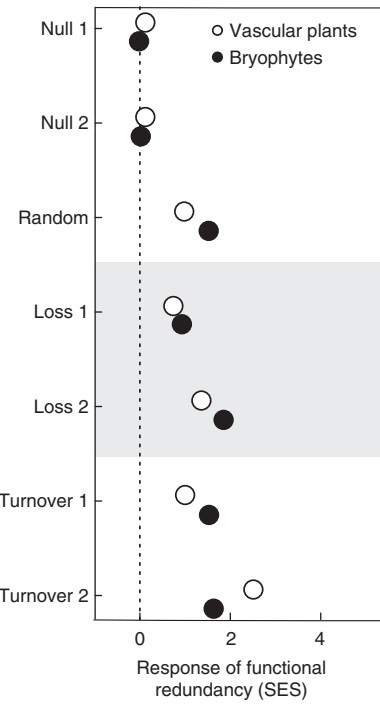

**Fig. 6** Effects of modelled species assembly scenarios on functional redundancy. The response of plant functional redundancy to seven different assembly scenarios (see text for a description) as compared to the observed functional redundancy, expressed as the deviation of the standardized effect sizes (SES) from zero

functional redundancy, suggesting that in this situation peat bogs would be less resilient and resistant to climate change[43]. The substitution of cluster-specific species by a random species from the other cluster also led to an erosion in functional redundancy at the community level. The results from our simulations are in contrast with the observations in our data set, where functional redundancy in European bogs along environmental gradients was conserved, suggesting a non-random compensatory species replacement in response to environmental change[44]. A corollary of our findings is that the functional roles of species lost are taken up by functionally identical species ('look-a-likes', ref. [18]). The fact that taxonomic and functional turnover in bog plant communities are decoupled suggests strong trait-based community assembly. Such decoupling may underlie the stability of functional redundancy; a mechanism that may strengthen the ability of peatlands to withstand environmental change.

Our analysis demonstrates the highly dynamic response of European peat bog plant species composition to changes in environmental conditions, with temperature and precipitation being the most important drivers, followed by atmospheric N and S deposition. We identified two plant species clusters, which were remarkably similar in functional composition, but showed distinct and opposite responses to environmental change. Despite relatively high taxonomic turnover, community level functional redundancy was maintained along the environmental gradients by opposing cluster-specific effects. Together these results suggest that taxonomic and functional turnover of peat bog plant communities along environmental gradients are decoupled. Importantly, a between-cluster replacement of functionally similar species seems to underlie such decoupling, and may moderate the effects of climate change on peat bog functioning. Our results have important implications for peatland conservation. Given environmental change is expected to continue or even intensify[45], European peat bog plant communities will change as species migrate to areas where environmental conditions are suitable. Our data suggest that peat bog functioning across the entire range in environmental conditions can be maintained if species migration is compensated by the arrival of functionally identical species. To facilitate this functional replacement, the conservation of the full European peat bog species pool, and investing in improving the connectivity of European peat bogs, should have highest priority.

random, and likely the result of a deterministic replacement of species with similar traits. Results from our community re-assembly simulations demonstrate that non-random species replacement underlies the apparent robustness of functional redundancy. Whilst it has been suggested that extreme environmental conditions, as present in peatlands, may select for a set of optimal trait values[42], these simulations highlight that apparent functional convergence of peatland species cannot merely explain our findings. Cluster–specific species loss resulted in a decrease in

## Methods

**Plant community composition.** In two consecutive summers (2010 and 2011), we collected abundance data for all vascular plant and bryophyte species from five randomly chosen hummocks and lawns (0.25 m$^2$ quadrats; 10 in total) across 56 European *Sphagnum*-dominated peatlands (Fig. 1; map source: ref. [46]). Vascular plants and *Sphagnum* mosses were identified to the species level. Non-*Sphagnum* bryophytes were identified to the family level. Lichens were recorded as one group. Identification to these taxonomic levels allowed us to include a larger number of sites in a more constrained time period. Following this taxonomic grouping, 59 taxa were included in the final data set (Supplementary Table 3). Species data were averaged for each peatland, resulting in site-level mean abundance values. To minimise the influence of rare species on further analyses, species occurring in less than five peatlands were not included in the final data set (Supplementary Table 3). Rarefaction curves (Supplementary Fig. 6a) indicated that our sampling adequately captured the species richness of European *Sphagnum*-dominated peatlands.

**Bioclimatic and atmospheric deposition data.** Four bioclimatic variables were extracted from the *WorldClim* database[47]: mean annual temperature, temperature seasonality, mean annual precipitation, and precipitation seasonality, and averaged over a 10 year (2000–2009) period. Moisture index was calculated as the ratio between mean precipitation and mean temperature in the warmest quarter (P: T$_{warm}$)[48]. Atmospheric deposition data were produced using the EMEP (European Monitoring and Evaluation Programme)-based IDEM (Integrated Deposition Model) model[49] and consisted of grid cell averages of total reduced (NH$_x$) and oxidised (NO$_y$) nitrogen and sulphur (SO$_x$) deposition. See Supplementary Table 1 for a full list and range values. Along the environmental gradients, spatial dependency (spatial autocorrelation) of the plant communities was tested using multiscale ordination analyses[50]. Species-environment relationships were not spatially structured, except for sites in close proximity (Supplementary Fig. 6b).

**Plant functional traits and community functional indices.** To calculate functional indices for the plant communities, we used trait data that were available for all the species across European peatlands. For the vascular plant species, we used data for a range of commonly measured and widely available traits from LEDA[51]: specific leaf area, canopy height, leaf dry matter content, seed mass and seed number. These data were complemented with data on life form and ecological strategy from BiolFlor[52] and Mycorrhizal association from MycoFlor[53]. For the bryophyte communities, we compiled trait data that considered their role in peatland function, as well as their capacity to compete. These traits included plant length[54], spore diameter and capsule diameter[55], productivity[56], tissue carbon, nitrogen and phosphorus content, stem width, length and width of hyaline cells, and length width of stem leaves[57] (Supplementary Table 4).

For each peatland, we calculated the functional identity, i.e., community-weighted mean (CWM) trait values, of the vascular plant and bryophyte communities, by multiplying the relative abundance of each species ($p_i$) by its trait value ($t_i$), then summing across the species present in the plant communities: CWM $= \sum p_i \times t_{ij}$, for species $i$ and trait $j$. Further, based on joint species distribution modelling (see below) we calculated co-response cluster-specific CWMs.

As a proxy for the resilience and resistance of peatland functions to environmental change, we calculated functional redundancy (FR). FR is based on the observation that some species perform similar roles in communities and ecosystems, and may therefore be substituted with little impact on ecosystem processes[58]. FR was defined as the difference between Simpson's species diversity (D) and functional diversity[59] (FD): FR = D-FD, and was calculated for the vascular plant and bryophyte communities separately. Functional diversity was based on the degree of functional dissimilarity between plant communities, taking differences in species abundance into account (Rao's quadratic entropy: RaoQ). FD was calculated using the 'FD' R package[60]. FR ranges from 0 to 1, where FR = 0 and FR = 1 indicate complete divergence or convergence, respectively, in traits between species. Functional turnover (functional β-diversity) between peatlands was then calculated as the change in Rao's quadratic entropy index, using the 'betaQmult' package in R[61]. To test the robustness of the values of our functional indices, and to test whether these values are affected by the number and identity of traits used in the calculations, we performed a random trait removal analysis. We compared observed numbers of functional diversity and redundancy with those of a set of scenarios where up to four (vascular plants) and eight (*Sphagnum* mosses) traits were removed randomly from the trait data set (always leaving four traits)[62]. Results from this analysis indicate that while variability increases with increased numbers of traits removed the overall effect of such removal is negligible (Supplementary Fig. 5).

**Data analyses.** As a first step in our data analysis, we used stepwise multiple regressions to identify the main bioclimatic and atmospheric deposition variables associated with patterns in species richness and diversity (Simpson's). We also analysed patterns in plant species turnover (β-diversity) along the environmental gradients, using generalised dissimilarity modelling[35] (GDM). GDM is widely used to identify the main environmental drivers for species turnover, and to test the independent significance of these drivers (using permutation tests). GDM is an

extension of Mantel correlation analysis using nonlinear regression[35]. The predictor matrix included the (non-correlated) bioclimatic and atmospheric deposition data, alongside the geographical distance (i.e., spatial distance; latitude, longitude) between peat bogs. These geographical indices provide insight into the amount of variation explained by the GDM that is attributable purely to geographical distance (i.e., space). For all variables the difference in deviance explained by the full model and a reduced model with the variable omitted from the model was calculated. If the difference exceeded 0.5%, we tested the significance of the variable using Monte Carlo permutation[35, 63]. Permutation testing overcomes the issue of data-dependence when using unfolded distance/dissimilarities, and thus, reduces the inferential uncertainty of GDM outcomes. Only significant variables were retained. From these significant variables, GDM calculates environmental dissimilarity (i.e., the scaled combination of inter-site distances based on all geographical and environmental variables) and plant compositional dissimilarity across sites, and returns I-spline coefficients highlighting the importance of each individual variable for species turnover. Additionally, we determined the influence of environmental dissimilarity on the functional dissimilarity between peatlands. To test the robustness and repeatability of the GDM outcomes, we compared four community-level models to GDM, covering a wide range of model-class types: (i) constrained additive ordination (CAO), (ii) constrained quadratic ordination (CQO), (iii) multiresponse multivariate adaptive regression splines (MMARS) and (iv) multiresponse multivariate artificial neural network (MMANN). All models were fitted with the same ten environmental variables used in GDM (i.e., we did not perform a priori variable selection). We first used 30% of the data (randomly selected) to tune parameter values, after which the best model was used with the full data[64]. From each model, we extracted the importance of each environmental variable in explaining species composition. Then, model outcomes were tested against GDM outputs using paired *t*-tests, and *t*-values converted into standardized effect sizes (SES) to express the difference between GDM and comparing model outcomes[65]. We defined the level of significant differences of SES by generating 1000 random model outputs (i.e., null importance of environmental variables) and compared them to GDM outputs using a similar procedure as before. All models were and analyses were run in R v.3.3.2, using codes and R packages described in ref. [64].

The main environmental variables returned from GDM were then used in joint species distribution modelling[66, 67] (JSDM) to identify patterns in the response of species. JSDM takes a hierarchical approach that combines species abundance and similarities in species responses to environmental variables. As such it can be used to assess species co-response to environmental variable. The model starts with a species joint distribution analysis (i.e., species co-occurrence) and builds on inverse prediction that quantifies how environmental variables affect a combined multivariate output, e.g., the distribution of co-occurring species[66, 67]. The resulting species correlations can then be decomposed into (i) correlations due to similar environmental responses and (ii) residual correlations (correlations between species that are not due to the environmental factors). JSDM was performed with the Markov Chain Monte Carlo Bayesian modelling software JAGS, using the *R2jags* package. We ran five chains of 10$^6$ iterations, with the first 15,000 discarded as burn-in. The remaining samples were thinned by a factor 1000, meaning that we retained 985 samples per chain for post-processing. We then built cluster diagrams representing either species co-occurrences due to shared environmental responses or residual species co-occurrences using the R package 'network'[68]. We considered species co-response to be robust when the correlation coefficient between species due to shared environmental response (or residual correlation) exceeded 0.6.

The next steps of our analyses aimed at understanding the relationships between species turnover and functional turnover. These analyses were performed for the vascular plant and bryophyte communities separately, as the key functional traits differ between these groups. To calculate species turnover we used Bray−Curtis resemblance coefficients[69], for comparability with GDM results. Functional turnover was calculated using a decomposition of the Rao quadratic entropy (Rao Q)[61]. The relationship between species composition and functional composition of vascular plant and bryophytes was tested using generalized linear models. We also tested the relationship between functional dissimilarity and environmental dissimilarity and how the main environmental variables affected functional redundancy (FR) of the vascular plant and bryophyte communities. The latter analysis was performed for the whole community, as well as for the co-response clusters separately.

Finally, the effects of directed and non-directed (i.e., random) species assembly, cluster-specific species loss, and between-cluster species replacement on FR were then tested using a series of simulations. We first used two null-models (ref. [70]) to compare FR-values obtained from 1000 null assemblages with FR from the original species assemblages. The first null model ('NULL1—occurrence re-assembly) comprises a random reshuffle of species occurrence whilst keeping the total site and species abundances intact. The second null model ('NULL2—abundance re-assembly) randomly assigns a new abundance value to only non-zero cells. NULL2 allows species abundance in each cell to change but preserves species occurrence and total site and species abundance in the matrix. Next, we tested the effect of species turnover on vascular plant and bryophyte community FR. First, we re-assembled plant communities by random selection of species from the original data set (RANDOM), calculated FR based on 1000 iterations, and compared FR with observed FR for the vascular plant and bryophyte communities. The number of species selected to build the RANDOM community equalled the average number

of species in these communities across all peatland sites; seven for vascular plants, five for bryophytes. Consequently, the RANDOM model did not preserve species occurrences nor total site and species abundances. As a second scenario we tested the effect of loss of species from either one of the previously described co-response clusters (LOSS1, cluster I loss; LOSS2, cluster II loss). Finally, we assessed the effect of between–cluster species replacement on the FR of the vascular plant and bryophyte communities. In the first scenario (TURNOVER1, I for II replacement), three species from cluster I were randomly replaced by the same number of species from cluster II. In the second scenario (TURNOVER2, II for I replacement), three species from cluster II were randomly replaced by the same number of species from cluster I. Again, observed FR for the vascular plant and bryophyte communities was compared with those obtained from 1000 matrices randomly re-assembled matrices. Standardized effect size (SES: $(FR_{observed} - FR_{simulated})/SD\_FR_{simulated}$) was then calculated and tested for departure from 'zero'. We used *base*, *vegan* and codes adapted from the *FD* and *EcosimR* R packages to perform these analyses.

**Data availability**. Plant community composition data, bioclimatic and atmospheric deposition data, and plant trait data can be accessed through the Dryad Digital Repository doi:10.5061/dryad.g1pk3. Plant trait data can further be accessed through the data repositories described in the paper. All codes used for statistical analyses are publically available through the R statistical environment. Specific codes can be obtained from the authors.

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

## Acknowledgements
The BiodivERsA-PEATBOG project was funded as an ERA-net project within the European Union's 6th Framework Programme for Research through NWO-ALW (832.09.003). The Dutch Foundation for the Conservation of Irish Bogs funded part of this study. We thank all agencies and landowners for peatland access, and all who assisted in the field. We are indebted to Jane Catford, Liesje Mommer, Heinjo During, Hans ter Steege and Hendrik Poorter for valuable comments. This paper is in memory of late Christian Blodau (1971–2016), an inspiring member of the PEATBOG team.

## Author contributions
Conceptualization and funding acquisition: L.B., S.J.M.C, N.B.D., B.H.S. and J.T.A.V.; Floristic data collection: B.J.M.R., L.B. and S.J.M.C, with help of M.M., R.J.P., N.B.D., K.Z. and J.T.A.V.; Bioclimatic resources: B.J.M.R., A.Bl. and R.J.P.; Trait data: B.J.M.R., V.E.J.J and J.K; Statistical analyses: B.J.M.R. and V.E.J.J.; Writing: B.J.M.R., V.E.J.J., R.J.P, L.B. A.Bu, J.v.R., and J.T.A.V. wrote the original draft, to which all authors contributed by review and editing.

## Additional information

**Competing interests:** All authors declare no competing financial interests.

