## [Peer Review File · Nature Communications]

Reviewers' comments:

Reviewer #1 (Remarks to the Author):

In this manuscript, the authors present data on plant communities collected in 56 peatlands across Europe. They aimed to establish relationships between plant species composition, trait composition, and some environmental drivers (mostly climatic). Species abundance were collected in the field while trait values were gathered from published databases. Their main findings is that despite plant species aggregate into two clusters that are each characterized by specific environmental conditions (significant taxonomic turnover), functional identity and redundancy remain similar between clusters (low functional turnover across peatlands). They authors concluded that plant taxonomic and functional turnover are decoupled in bogs, which made these ecosystems resilient to future environmental change.

The main strength of the study, is the extensive field works that was done, that allow the study of species turnover at a continental scale. Statistical analyzes seem adequate and take into account their main methodological limitations.

Main comments:

1) The results of the study are undoubtedly of great interest to science, although they are not so surprising in themselves. Peatlands are indeed ecosystems where local environmental constraints are so strong (e.g., acidity, waterlogged conditions) that only some species with specific functional traits can thrive. On this subject, I am surprised that the authors do not present any information on the local environmental conditions of their sampling sites (eg, water pH and conductivity, water table level), conditions that strongly influence the floristic composition of peatlands (data should be presented at least as supplementary files). The authors indicate that they sampled only Sphagnum peatlands (bogs), but their list of species suggests that some sites were richer than others. Indeed, some species that are often associated with more minerotrophic than ombrotrophic conditions (at least in North America) were sampled such as *Menyanthes trifoliata* and *Trichophorum cespitosum*.

2) The authors seem also to consider that taxonomic and functional diversity can only follow the same pattern (e.g., Lines 55-56). However, the relationship between taxonomic and functional diversity has been previously shown to vary greatly depending of several factors such as number and type of traits included in the study and species richness (see some references below). This aspect should be at least discuss in the manuscript. For example, does the fact that only 4 traits have been used for vascular plants versus 12 for Sphagnum mosses may explain that the relationship between functional and taxonomic diversity appears to be higher for Sphagnum moss than for vascular plants (Figure 3B).

Naaf & Wulf, 2012; Does taxonomic homogenization imply functional homogenization in temperate forest herb layer communities? *Plant Ecology*, 213, 431–443.

Smart S.M., Thompson K., Marrs R.H., Le Duc M.G., Maskell L.C., & Firbank L.G. (2006) Biotic homogenization and changes in species diversity across human-modified ecosystems. *Proceedings. Biological sciences* 273, 2659–2665.

Sonnier G., Johnson S.E., Amatangelo K.L., Rogers D. a., & Waller D.M. (2014) Is taxonomic homogenization linked to functional homogenization in temperate forests? *Global Ecology and Biogeography*, 23, 894–902

Baiser B. & Lockwood J.L. (2011) The relationship between functional and taxonomic homogenization. *Global Ecology and Biogeography*, 20, 134–144.

Fukami T., Bezemer T.M., Mortimer S.R., & Van Der Putten W.H. (2005) Species divergence and trait convergence in experimental plant community assembly. *Ecology Letters*, 8, 1283–1290

Minor comments:

- 1) The graphic quality of most figures is relatively low, making their understanding difficult. I imagine that it is only a problem of transformation towards the pdf, but if not the authors would have to solve the problem.
- 2) line 133: insert "species" between bog and such.
- 3) There is 59 species in Table 2 and Table S2, but at line 275 the authors refers to 61 species.
- 4) Lines 166-167, should it be Figures 3D instead of 3C?
- 5) Line 279: Figure 5A instead of 6a
- 6) Line 292: Figure 5B instead of 6B
- 7) Lines 294-303: Using traits from trait databases can heavily under or overestimate regional trait characteristics (e.g., Cordlandwehr et al. 2013, *J. Ecol.* 101). Please discuss somewhere the implications of it.

Reviewer #2 (Remarks to the Author):

Thank you for this interesting study on spatial variation in taxonomic and functional trait composition of peat bog plant communities. I feel the issues raised during the previous round of review have been adequately addressed, though I do have a few addition issues to raise.

MAJOR

(1) The basic thesis of this study is that functional redundancy of plant taxa along environmental gradients may buffer peat bogs from loss of ecosystem function should species composition shift in response to global change. While I generally agree with this logic, it requires the assumption that changes in plant species composition in response to future global change drivers will occur via replacement of functionally similar species from within the broader species pool of peat bog taxa. However, no evidence / references are provided to support this unstated assumption. Addressing this concern may be as simple as stating that given the unique environments (hydrology, nutrient status, etc) of peat bogs, only plant species adapted to such conditions could be expected to colonize these habitats & given the findings, are expected to have redundant functional traits. Of course, should the magnitude of environmental change become large enough, then this assumption presumably becomes less and less tenable.

(2) Figure 1C shows the fitted splines from GDM. While it true that these plots indicate the relative importance of each variable in explaining species turnover (and the total amount of compositional turnover associated with that variable), it is not necessarily true that they also represent the "proportion of total explained deviance attributable purely to each" environmental variable. I think this plot is useful, but I suggest the authors drop mention of deviance for 1C and consider including an additional panel that plots variable importance as the percent reduction in deviance explained between models fit with the permuted / un-permuted variable, which is a better indicator of explanatory power (and was done for the comparisons with other JSDMs as described on lines 330-333).

MINOR

P2, L38: suggest change to "how ecosystem function"

P3, L64-65: suggest delete "strikingly" and "highly"

P5, L133: Missing word? "bog PLANTS such as"

P6, L159: suggest change "poorly" to "weakly"

P8, L225-244: suggest use of "environmental gradients" instead of "the environmental gradient(s)". In other words, drop "the" and make "gradient" plural throughout.

P11, L348: Strictly speaking, these other models do not use species turnover as the response, but rather composition. In other words, GDM models a distance, but the other methods model species composition itself. Suggest change "species turnover" to "species composition".

Figure 1C - What are the units of these variables? Are some of the axes log-scale?

Reviewer #1 (Remarks to the Author):

In this manuscript, the authors present data on plant communities collected in 56 peatlands across Europe. They aimed to establish relationships between plant species composition, trait composition, and some environmental drivers (mostly climatic). Species abundance were collected in the field while trait values were gathered from published databases. Their main findings is that despite plant species aggregate into two clusters that are each characterized by specific environmental conditions (significant taxonomic turnover), functional identity and redundancy remain similar between clusters (low functional turnover across peatlands). They authors concluded that plant taxonomic and functional turnover are decoupled in bogs, which made these ecosystems resilient to future environmental change.

The main strength of the study, is the extensive field works that was done, that allow the study of species turnover at a continental scale. Statistical analyzes seem adequate and take into account their main methodological limitations.

Main comments:

1) The results of the study are undoubtedly of great interest to science, although they are not so surprising in themselves. Peatlands are indeed ecosystems where local environmental constraints are so strong (e.g., acidity, waterlogged conditions) that only some species with specific functional traits can thrive. On this subject, I am surprised that the authors do not present any information on the local environmental conditions of their sampling sites (eg, water pH and conductivity, water table level), conditions that strongly influence the floristic composition of peatlands (data should be presented at least as supplementary files).

– We fully agree with the referee that peatlands are clearly constrained by environmental conditions, such as low pH and waterlogged conditions. However, we do not think this makes our results less surprising. We show that within these ecosystems, climatic variation can lead to substantial taxonomic turnover. Although all these species will to some extent be adapted to the environmental constraints in these systems, they do show considerable variation in functional traits (see Table 1). Nevertheless, functional identity and redundancy remained largely unchanged. We think this is an important finding.

Table 1 | Range of specific leaf area (SLA), canopy height, and leaf dry matter content –three important traits– in our data, as compared to the range (2.5 – 97.5% quantiles) as present in the TRY database: Kattge *et al.* TRY – a global database of plant traits. *Glob. Chang. Biol.* **17**, 2905–2935 (2011). Apart from canopy height, which includes trees in the TRY database and not in ours, the range of trait values in our study is consistent with the range in trait values as presented in the TRY database.

Trait	This study		TRY database	
	Min	Max	2.5% Quantile	97.5% Quantile
SLA (cm ² cm ⁻¹)	7.6	49.4	4.5	47.7
Canopy height (m)	0.06	0.85	0.04	30
LDMC (mg g ⁻¹)	124.1	465.5	100	420

That said, we agree with the reviewer that in general, it is important to provide information about important environmental conditions of study sites. Given the very large geographic scope of our study, however, the sites could not be visited on multiple occasions to measure these conditions. This is important because many of these conditions may vary considerably, both spatially and temporally. For example, during the sampling campaign we witnessed a site change from relatively dry to soaking wet due to a heavy rain event, affecting water quality data. At the opposite end, summer drought led to increased nutrient concentrations in pore water measured after extractions. As such, we are reluctant to present these data as accurate site descriptions. Instead, we used modelled long-term averages of the key climatic variables, which ultimately drive these local conditions. We are

convinced that with the data that we currently have, this is the most reliable and robust analytical approach.

The authors indicate that they sampled only Sphagnum peatlands (bogs), but their list of species suggests that some sites were richer than others. Indeed, some species that are often associated with more minerotrophic than ombrotrophic conditions (at least in North America) were sampled such as Menyanthes trifoliata and Trichophorum cespitosum.

– Indeed, we sampled *Sphagnum*-dominated peatlands. The referee is correct in noting that some species, such as *Menyanthes trifoliata* and *Trichophorum cespitosum* are commonly found in minerotrophic conditions. This does not mean, however, that these species do not occur ombrotrophic bogs. For example, both species are abundant in blanket bogs in the UK, which are commonly considered ombrotrophic. In addition, it is important to note that the classification of our study sites was based on the composition of the plant community as a whole, not on the occurrence of one or two species.

2) The authors seem also to consider that taxonomic and functional diversity can only follow the same pattern (e.g., Lines 55-56). However, the relationship between taxonomic and functional diversity has been previously shown to vary greatly depending of several factors such as number and type of traits included in the study and species richness (see some references below). This aspect should be at least discuss in the manuscript. For example, does the fact that only 4 traits have been used for vascular plants versus 12 for Sphagnum mosses may explain that the relationship between functional and taxonomic diversity appears to be higher for Sphagnum moss than for vascular plants (Figure 3B).

Naaf & Wulf, 2012; Does taxonomic homogenization imply functional homogenization in temperate forest herb layer communities? *Plant Ecology*, 213, 431–443.

Smart S.M., Thompson K., Marrs R.H., Le Duc M.G., Maskell L.C., & Firbank L.G. (2006) Biotic homogenization and changes in species diversity across human-modified ecosystems. *Proceedings. Biological sciences* 273, 2659–2665.

Sonnier G., Johnson S.E., Amatangelo K.L., Rogers D. a., & Waller D.M. (2014) Is taxonomic homogenization linked to functional homogenization in temperate forests? *Global Ecology and Biogeography*, 23, 894–902

Baiser B. & Lockwood J.L. (2011) The relationship between functional and taxonomic homogenization. *Global Ecology and Biogeography*, 20, 134–144.

Fukami T., Bezemer T.M., Mortimer S.R., & Van Der Putten W.H. (2005) Species divergence and trait convergence in experimental plant community assembly. *Ecology Letters*, 8, 1283–1290

– The reviewer raises an important point; functional trait indices can indeed vary, depending on the scale and focus. We address this issue in the revised manuscript (p.8) based on the references pointed to by this referee. According to recent findings, however, the number of traits we use in our analyses (8 for vascular plants – not 4 as the referee suggests – and 12 for *Sphagnum* mosses) exceeds the suggested number of traits required for reliable calculations (Maire et al. 2015; *Global Ecology and Biogeography*, 24(6), 728–740. <http://doi.org/10.1111/geb.12299>). Nonetheless, we recognize that the influence of the number of traits and their type may be a concern. To investigate this, we analysed the robustness of our approach, following Maire *et al.* (2015). First, we randomly removed 1 to 4 plant traits in the case of vascular plants, and 1 to 8 traits for *Sphagnum*, and then calculated functional redundancy and diversity for each of these scenarios (i.e. 1 trait removed, 2 traits removed, etc.) based on 1000 iterations. The functional indices of these new traits compositions were then compared to the observed indices using standardized effect sizes (SES). These results (Supplementary Fig. 5) show that there is no effect of random trait removal until only four traits are left to calculate the functional indices. Therefore, we think that although the reviewer raises a valid point, this is not a critical issue in our study. In revising the manuscript, we have included discussion of this point (p. 8), presented a new supplementary figure showing the results referred to above, and added the details of these analyses to the Methods section (p. 11).

Minor comments:

1) The graphic quality of most figures is relatively low, making their understanding difficult. I imagine that it is only a problem of transformation towards the pdf, but if not the authors would have to solve the problem.

– We regret the referee had some issues with the quality of the figures. We did upload our ‘high quality .eps-files’ as well, but transformation to the ‘.pdf-file’ may have compromised their quality. We hope this issue will be solved in this version, or that the referee may be able to download the high resolution files.

2) *line 133: insert "species" between bog and such.*

– We regret this incompleteness in the sentence, which is now fixed in the new version.

3) *There is 59 species in Table 2 and Table S2, but at line 275 the authors refers to 61 species.*

– We assume here that the referee is referring to Supplementary Table 3, not 2. The referee is absolutely right about the number of taxa used in our analyses. We corrected the number in the revised manuscript. As a corollary of the referees’ comment we slightly adjusted the caption of Supplementary Table 3, and included a missing species in the footnote of Supplementary Table 2.

4) *Lines 166-167, should it be Figures 3D instead of 3C?*

– We apologize for the incorrect referencing, which we have now corrected throughout the whole ‘Functional turnover and redundancy’ paragraph.

5) *Line 279: Figure 5A instead of 6a*

6) *Line 292: Figure 5B instead of 6B*

– We appreciate the attentiveness of the referee. Due to the addition of a new Supplementary Figure, based on our response to the 2nd comment of this referee, the reference to this graph is now correct.

7) *Lines 294-303: Using traits from trait databases can heavily under or overestimate regional trait characteristics (e.g., Cordlandwehr et al. 2013, J. Ecol. 101). Please discuss somewhere the implications of it.*

– In revising the manuscript we have briefly discussed this issue but also added that in the same paper, the authors conclude that average trait values to be unproblematic.

Reviewer #2 (Remarks to the Author):

Thank you for this interesting study on spatial variation in taxonomic and functional trait composition of peat bog plant communities. I feel the issues raised during the previous round of review have been adequately addressed, though I do have a few addition issues to raise.

MAJOR

(1) The basic thesis of this study is that functional redundancy of plant taxa along environmental gradients may buffer peat bogs from loss of ecosystem function should species composition shift in response to global change. While I generally agree with this logic, it requires the assumption that changes in plant species composition in response to future global change drivers will occur via replacement of functionally similar species from within the broader species pool of peat bog taxa. However, no evidence / references are provided to support this unstated assumption. Addressing this concern may be as simple as stating that given the unique environments (hydrology, nutrient status, etc) of peat bogs, only plant species adapted to such conditions could be expected to colonize these habitats & given the findings, are expected to have redundant functional traits. Of course, should the magnitude of environmental change become large enough, then this assumption presumably becomes less and less tenable.

– We agree with the reviewer that our study does assume that replacement of species will be within the pool of bog species. In this respect the large geographic range of our study is an asset; our sampling includes peatlands from throughout the European bioclimatic range and therefore samples most peatland species.

The reviewer refers to extreme climatic change, but only in unrealistically drastic climate change leading to total peat oxidation would peatland species be likely to be replaced by those of

mineral soils. To avoid ambiguity in our intention we added some wording on the fact that our proposed mechanisms works within the studied environmental gradient: “Our data provide empirical evidence that, within the studied environmental constraints in which peatland plants can exist, peatland plant community taxonomic and functional composition are decoupled.”

Further, we highlighted that our species replacement analysis rules out trait convergence amongst peatland species: “Whilst it has been suggested that extreme environmental conditions, as present in peatlands, may select for a set of optimal trait values , these simulations highlight that apparent functional convergence of peatland species cannot merely explain our findings.” (p. 8).

(2) Figure 1C shows the fitted splines from GDM. While it true that these plots indicate the relative importance of each variable in explaining species turnover (and the total amount of compositional turnover associated with that variable), it is not necessarily true that they also represent the "proportion of total explained deviance attributable purely to each" environmental variable. I think this plot is useful, but I suggest the authors drop mention of deviance for 1C and consider including an additional panel that plots variable importance as the percent reduction in deviance explained between models fit with the permuted / un-permuted variable, which is a better indicator of explanatory power (and was done for the comparisons with other JSDMs as described on lines 330-333).

– We agree with the reviewer. We followed the reviewers’ advice and calculated the percentage of deviance explained by each variable added in the GDM model. Hence, we added a panel to Figure 1 showing the percentage of deviance explained for all variables used in the GDM. This analysis highlights MAT, TS, MAP, SOx and NHx as the main explanatory variables underlying taxonomic turnover.

MINOR

P2, L38: suggest change to "how ecosystem function"

– Changed as suggested.

P3, L64-65: suggest delete "strikingly" and "highly"

– We followed the referees’ suggestion to omit these words.

P5, L133: Missing word? "bog PLANTS such as"

– Following the advice of ref. 1, we added the word ‘species’.

P6, L159: suggest change "poorly" to "weakly"

– Changed as suggested.

P8, L225-244: suggest use of "environmental gradients" instead of "the environmental gradient(s)". In other words, drop "the" and make "gradient" plural throughout.

– We followed this suggestion, as it refers to a more general pattern; exactly what we want to convey.

P11, L348: Strictly speaking, these other models do not use species turnover as the response, but rather composition. In other words, GDM models a distance, but the other methods model species composition itself. Suggest change "species turnover" to "species composition".

– The referee is absolutely right. We changed the wording according to this recommendation.

Figure 1C - What are the units of these variables? Are some of the axes log-scale?

– These are all partial distances, hence dimensionless.

REVIEWERS' COMMENTS:

Reviewer #1 (Remarks to the Author):

I have read carefully the manuscript revised by the authors as well as their reply letter and I consider that the authors have responded adequately to my various comments. I have no additional corrections to suggest and congratulates the authors for their work.

Reviewer #2 (Remarks to the Author):

I have reviewed the revised manuscript and the letter describing the response to the reviews. I feel all remaining issues have been satisfactorily addressed.

We appreciate the contribution of both referees on an earlier version of our manuscript, which greatly help to improve the quality of our manuscript. We a very happy to see both referees appreciate our revision, underlined by the absence of further issues or corrections.

Reviewer #1 (Remarks to the Author):

I have read carefully the manuscript revised by the authors as well as their reply letter and I consider that the authors have responded adequately to my various comments. I have no additional corrections to suggest and congratulates the authors for their work.

Reviewer #2 (Remarks to the Author):

I have reviewed the revised manuscript and the letter describing the response to the reviews. I feel all remaining issues have been satisfactorily addressed.